# A systematic review of neurological impairments in myalgic encephalomyelitis/ chronic fatigue syndrome using neuroimaging techniques

Rebekah Maksoud[1,2]*, Stanley du Preez[1,2,3,4], Natalie Eaton-Fitch[1,2,3], Kiran Thapaliya[1,2], Leighton Barnden[1,2], Hélène Cabanas[1,2], Donald Staines[1,2], Sonya Marshall-Gradisnik[1,2]

1 National Centre for Neuroimmunology and Emerging Diseases, Menzies Health Institute Queensland, Griffith University, Gold Coast, Australia, 2 Consortium Health International for Myalgic Encephalomyelitis, Griffith University, Gold Coast, Australia, 3 School of Medical Sciences, Griffith University, Gold Coast, Australia, 4 School of Medicine, Griffith University, Gold Coast, Australia

* ncned@griffith.edu.au

**Data Availability Statement:** All relevant data are within the paper and its Supporting Information files.

## Abstract

### Background

Myalgic encephalomyelitis/ Chronic Fatigue Syndrome (ME/CFS) is a multi-system illness characterised by a diverse range of debilitating symptoms including autonomic and cognitive dysfunction. The pathomechanism remains elusive, however, neurological and cognitive aberrations are consistently described. This systematic review is the first to collect and appraise the literature related to the structural and functional neurological changes in ME/ CFS patients as measured by neuroimaging techniques and to investigate how these changes may influence onset, symptom presentation and severity of the illness.

### Methods

A systematic search of databases Pubmed, Embase, MEDLINE (via EBSCOhost) and Web of Science (via Clarivate Analytics) was performed for articles dating between December 1994 and August 2019. Included publications report on neurological differences in ME/CFS patients compared with healthy controls identified using neuroimaging techniques such as magnetic resonance imaging, positron emission tomography and electroencephalography. Article selection was further refined based on specific inclusion and exclusion criteria. A quality assessment of included publications was completed using the Joanna Briggs Institute checklist.

### Results

A total of 55 studies were included in this review. All papers assessed neurological or cognitive differences in adult ME/CFS patients compared with healthy controls using neuroimaging techniques. The outcomes from the articles include changes in gray and white matter volumes, cerebral blood flow, brain structure, sleep, EEG activity, functional connectivity

**Funding:** This research was supported by the Stafford Fox Medical Research Foundation, the Mason Foundation, Mr. Douglas Stutt, Blake Beckett Foundation, Alison Hunter Memorial Foundation, the McCusker Charitable Foundation, Buxton Foundation, Mr and Mrs Stewart, Henty Community, Henty Lions Club and the Change for ME Charity. The funders had no role in study design, data collection and analysis, decision to publish, or preparation of the manuscript.

**Competing interests:** The authors have declared that no competing interests exist.

and cognitive function. Secondary measures including symptom severity were also reported in most studies.

## Conclusions

The results suggest widespread disruption of the autonomic nervous system network including morphological changes, white matter abnormalities and aberrations in functional connectivity. However, these findings are not consistent across studies and the origins of these anomalies remain unknown. Future studies are required confirm the potential neurological contribution to the pathology of ME/CFS.

## Background

Myalgic Encephalomyelitis/ Chronic Fatigue Syndrome (ME/CFS) is a complex spectrum disorder that affects 0.2–2.6% of the global population [1,2]. ME/CFS patients present with an array of symptoms including cognitive, autonomic and neuroimmune disturbances, endocrine dysfunction, in addition to impaired cellular energy metabolism and ion transport [3]. The aetiology and pathomechanism underlying ME/CFS remains elusive [4]. Moreover, there are no accepted targeted treatment regimens or diagnostic tests available [5]. Diagnosis is instead achieved through the application of symptom-specific case criteria when all other explanatory clinical causes have been excluded [6–8].

There are three criteria widely used to define ME/CFS patients: Fukuda (1994), Canadian Consensus Criteria (2003) (CCC) and the International Consensus Criteria (2011) (ICC) [6–8]. The Fukuda criteria, published by the Center of Disease Control and Prevention (CDC) places emphasis on persistent fatigue that is unrelieved by rest in combination with at least four out of a potential eight additional symptoms, including but not limited to joint pain, throat soreness and impaired memory function [6]. The CCC and ICC built upon the Fukuda criteria and included important hallmark symptoms including post-exertional malaise, immunological and neurological symptoms that were not accounted for in the original Fukuda definition [7,8]. Although there is no consensus on the pathomechanism of ME/CFS, it has been classified as a neurological disease by the World Health Organization since 1969 as autonomic and cognitive dysfunction are key underlying features of this illness [9].

The nervous system is a tightly integrated network of nerves and specialised neuronal cells that orchestrates many critical physiological functions including motor, sensory and cognitive processing [10]. Evidence of disruption to this network is a consistent feature of ME/CFS that has been assessed through various neuroimaging techniques [11–48]. Neuroimaging studies involving the use of tools including but not limited to magnetic resonance imaging (MRI), positron emission tomography (PET) and electroencephalography (EEG) have enabled advancements in the detection of structural and functional abnormalities in ME/CFS and other neurological diseases.

The primary aim of this systematic review was to collect and appraise the literature related to the structural and functional neurological changes in ME/CFS patients as measured by imaging techniques. Secondary to this, how neurological changes may influence onset, symptom presentation and severity of the illness was also investigated. Cognitive impairment, sleep and energy wave impairments, functional connectivity (FC), gray and white matter (WM) changes and cerebral blood flow in adult ME/CFS patients compared with healthy controls (HCs) form the major focal points of this review. This research serves as a platform to evaluate

the usefulness of these imaging processes in characterising ME/CFS that may translate into a potential future diagnostic tool and targets for pharmacotherapeutic intervention.

## Literature search

This review was conducted in accordance with the Cochrane reviews and Preferred Reporting Items for Systematic Reviews and Meta-Analyses (PRISMA) guidelines (Fig 1). Relevant literature was retrieved through databases Pubmed, Embase, MEDLINE (via EBSCOhost) and Web of Science (via Clarivate Analytics). A systematic search on these databases of full- text and Medical Subject Headings (MeSH) terms "Syndrome, Chronic Fatigue" along with "Functional Neuroimaging" or "Diagnostic Imaging" or "Tomography, X-Ray Computed" or "Magnetic Resonance Imaging" or "Magnetic Resonance Spectroscopy" or "Magnetoencephalography" or "Positron-Emission Tomography" or "Tomography, Emission-Computed, Single-Photon" or "Electroencephalography" (S1 File) was performed between December 1994 and August 2019. Search terms were combined using Boolean operators 'OR' to expand the search for all expressions of cases and 'AND' to specify cases containing Syndrome, Chronic Fatigue in conjunction with the listed neuroimaging terms. Two literature searches were completed for this systematic review on separate occasions by two authors using the same search method. The primary search was conducted on the 19th of August 2019 by RM and the 2nd search was conducted on the 12th of September 2019 by SDP using the same methodological approach. Reference list checking and citation searching was completed, and no additional papers were selected. Searching for unpublished literature was not performed. No additional papers were identified in the final search or through alternative databases such as Griffith University institute library or Google Scholar.

## Inclusion and exclusion criteria

Studies that contained two or more key search terms in the abstract or title and adhered to the following inclusion criteria were selected for review: (i) published in 1994 or later; (ii) conducted in human adults aged 18 years or over; (iii) written in English and available as full text; (iv) were journal articles reporting on original research; (v) diagnosis of ME/CFS used the following case criteria: Fukuda (1994), CCC (2003), ICC (2011) or the Institute of Medicine (IOM) diagnostic criteria (2015); (vi) all studies investigated neurological changes in ME/CFS patients using neuroimaging techniques compared to HCs.

Articles were excluded from this review if they did not include at least two key search terms in the abstract or title or if they adhered to any of the following exclusion criteria: (i) written prior to the establishment of the Fukuda definition in 1994; (ii) conducted in human participants that were under 18; (iii) not written in English or not available as full text; (iv) were studies reporting on non-original data including: duplicate studies, case reports or review articles; (v) use of alternative case criteria than those aforementioned; (vi) studies were not relevant to the scope of this review. Publications were also excluded if the ME/CFS cohort was compared with another patient group (e.g., fibromyalgia, or depression or chronic fatigue etc.) and not compared with HC. All interventional studies were excluded.

## Selection of studies

The open-source reference management tool Zotero was used to screen, sort and store the retrieved articles from all databases. Screening involved reviewing and selecting articles based on eligibility and exclusion criteria. Publications were selected independently by two different authors using the same protocol to ensure the validity of the search. After selection of all

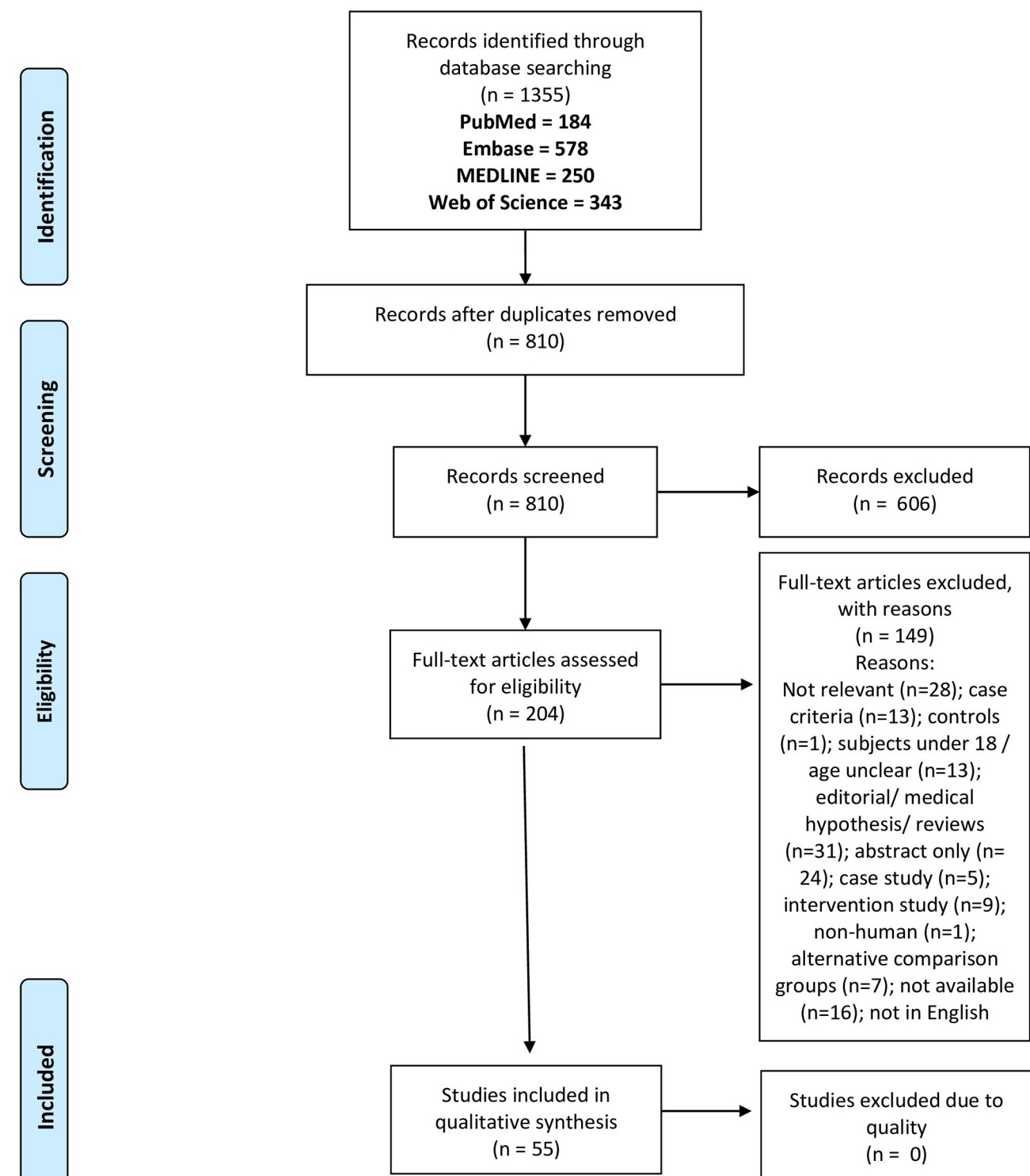

**Fig 1. PRISMA flow diagram of literature search for included studies in this review of neuroimaging and ME/CFS.**

papers included in this review, the relevance of these papers was reassessed by other co-authors.

### Data extraction

Following the selection of papers relevant data was extracted including (i) study design; (ii) diagnostic criteria used; (iii) sample size; (iv) method of analysis/ neuroimaging technique used.

### Quality assessment

All publications included in this systematic review were evaluated for quality and bias using the Joanna Briggs Institute (JBI) Checklist (S2 File). Quality assessment was completed by two authors (RM and SDP) on separate occasions. Each checklist item assesses the following: (1) group matching, (2) source population, (3) criteria, (4) method of exposure, (5) assessment of exposure, (6) identification of confounding variables, (7) management of confounding variables, (8) measurement of outcomes, (9) exposure period selection, (10) statistical analysis. Items 4, 5 and 9 were excluded in selected papers as these items are specific for intervention studies.

## Results

Across all databases, Pubmed (184), Embase (578), MEDLINE (250) and Web of Science (343), a total of 1355 papers were identified using the selected search terms. Following removal of all duplicates and application of all exclusion and inclusion criteria the total number of articles was refined to 55. The selection process as conducted according to the PRISMA guidelines has been outlined in Fig 1. All papers assessed neurological differences in adult ME/CFS patients compared to HCs using neuroimaging techniques.

### Participant and study characteristics

Participant and study characteristics are presented in (S3 and S4 Files). Out of the 55 articles included in this review, 52 were observational case-control studies [11–18,49–51] and three (6.25%) were observational twin studies. The mean number of ME/CFS patients and HCs included across all studies were 22.9 and 19.3 respectively. Females made up the largest proportion of participants. There were only 10 studies which reported race, where majority of the participants were Caucasian [12,19,21,26,27,37,41,47,48,52]. Mean age across all studies was 40.9 years for ME/CFS patients and 39.07 years for HCs. Only 16 out of 55 papers reported a value for body mass index (BMI) or weight [11,12,17–19,35,37,38,41,43,53–55]. In all twin studies, levels of zygosity were assessed and ascertained at 99.9% [12,21,27]. 47 of the 55 identified papers included participants that fulfilled the requirements of being diagnosed with the Fukuda criteria [12,14,16–19,21–28,30,31,34–41,43–61]. Three of the papers used the more stringent CCC criteria to classify ME/CFS patients [15,29,42]. The remaining five research papers used either of the two aforementioned criteria [11,13,20,33,62]. No papers utilised the IOM diagnostic criteria. For ME/CFS patients, mean illness duration was 9.68 years.

There were four different neuroimaging techniques used to assess neural changes in ME/CFS compared with HCs. Out of the 55 studies: 16 studies utilised MRI [11,13–15,20,31–36,43,53,58,62,63], 17 used functional MRI (fMRI) [16,18,21–24,26,34,40–43,45,49,49–51], five used PET scans [17,25,28,30,44] and 11 used EEG [12,27,29,38,39,46–48,54,55]. The remaining studies used magnetic resonance spectrometry (MRS) [19,52,59,60,64,65]. The articles reported on multiple outcomes including brain structure, cerebral blood flow (CBF), gray and WM volumes, sleep and EEG, and FC. Key findings of each of the included studies have been described in (S5 File).

## Literature reporting changes in cerebral blood flow

Of the included studies, six investigated changes in cerebral blood flow (CBF) in ME/CFS patients compared with HCs [20,21,49,51,62]. Two of the studies reported significantly lower CBF in ME/CFS patients compared with HC [19,62]. A study that utilised arterial spin labelling (ASL) saw less regional (rCBF) in the left cingulate cortex and right lingual gyrus in ME/CFS patients [19]. This study also reported higher ventricular lactate in ME/CFS patients correlated with several measures of physical health and disability using MRS [19]. Another ASL MRI study saw a significantly less global CBF in nine of the 11 patients studied, however the remaining two ME/CFS patients showed greater CBF relative to HCs which did not reach significance [49]. Another study showed that there were significantly higher CBF levels when provided with a Paced Auditory Serial Addition Test (PASAT) cognitive task, this then decreased after three minutes. ME/CFS patients had significantly lower rCBF in superior temporal gyri (STG), precuneus and fusiform gyrus during recovery compared to HCs [20]. A single-photon emission computed tomography (SPECT) twin study found that there were no significant differences in rCBF values between a healthy twin and a twin discordant for ME/CFS [21]. In an ASL study no group differences were found in CBF or heart rate variability (HRV), however, there was an inverse correlation between CBF, HRV and fatigue symptoms [51].

An ASL study investigated orthostatic intolerance (OI) in ME/CFS patients compared with HCs. Severity of OI symptoms was associated with lower intracranial compliance (change in volume per unit change in pressure), and higher intracranial perfusion. There were no significant differences in intracranial compliance or perfusion between ME/CFS patients and HCs [53].

## Literature reporting changes in brain structure and function

One of the most frequently reported structural or functional differences occurred in the cingulate region. This feature was described in 15 studies [18,19,22–30,44,50,57,62]. A PET study reported a 199% greater binding potential of 1C-R-PK11195, which shows neuroinflammation, in the cingulate region of ME/CFS patients compared with HCs. There was also a significantly greater binding potential in the hippocampus, amygdala, thalamus, midbrain and pons of patients compared with HC. For the cingulate region and the thalamus, this difference positively correlated with pain scores while the binding potential in the hippocampus correlated with the depression score [25]. Another PET study found that there was significantly less [$^{11}$C] (+) MCN5652, a radiotracer used to detect serotonin transporters (5-HTT), in the anterior cingulate of ME/CFS patients indicating a smaller density of 5-HTT [28]. There were also reports of higher FC between the posterior cingulate cortex and the dorsal anterior cingulate cortex in female ME/CFS patients that had strong positive association with the Chalder Fatigue Scale score [24]. This finding was corroborated by two other studies, one using the same method and another using ASL [16,62]. When fatigue was induced in ME/CFS patients, a BOLD fMRI study found a significantly higher activation of the posterior cingulate gyrus, occipito-parietal cortex and para-hippocampal gyrus in ME/CFS patients compared with HCs, while dorsolateral and dorsomedial prefrontal cortex activation were lower. In a fluorodeoxyglucose (FDG)-PET study 12 patients showed bilateral hypometabolism of glucose in the cingulate gyrus and adjacent mesial cortical areas compared to HCs while two patients showed hypometabolism in the cuneus/ precuneus [44]. In ME/CFS patients, a significantly higher perfusion of the anterior cingulate region and less perfusion elsewhere was exhibited in an investigation using SPECT [26].

Disruption of hippocampus function was described in six studies [16,22,25,27,50,58]. Significant differences in current source density was found in the parahippocampal gyrus in ME/CFS patients compared with HCs [27]; this includes higher EEG delta power. Significantly

reduced FC was also reported in the left parahippocampal gyrus of ME/CFS patients [16]. When dynamic FC (dFC) was assessed it was also found that it was reduced between the hippocampus and right superior parietal lobe in ME/CFS patients and this correlated with task-related fatigue [50]. BOLD fMRI studies found that para-hippocampal gyrus activation was higher in patient groups during a fatigue inducing task [23]. Significantly reduced serotonin receptors were detected using PET in the hippocampus of ME/CFS patients compared with HC [17]. The ligand 1C-(R)-PK11195 was increased in the hippocampus of ME/CFS patients signifying increased neuroinflammation and this correlated with depression scores [25]. Significantly lower gray matter volume (GMV) was also detected in the posterior division of the left parahippocampal gyrus in ME/CFS patients compared with HCs [58].

T1 weighted spin echo (T1wSE) and T2 weighted spin echo (T2wSE) studies found significant increases in T1wSE, which responds to myelin levels in WM, in prefrontal WM (13) and sensorimotor WM [15]. One of these studies also reported significant abnormal autonomic correlations in the brainstem vasomotor centre, midbrain reticular formation and hypothalamus which were interpreted to result from impaired connections between them [14]. The other study showed a strong link between brainstem GMV and pulse pressure in ME/CFS patients compared with HCs [11].

While performing the valsalva maneuver ME/CFS patients that test positive or negative for a temporomandibular disorder (TMD) and HCs all show activation of the superior and inferior frontal gyri, the left and right putamen and thalamus, and the insular cortex. Compared with those who screened negative for the TMDs and HCs, ME/CFS patients that have an accompanying TMD showed greater activity in the left insular cortex and left caudate nucleus [61].

There were eight papers in total that discussed GMV in ME/CFS patients compared with HCs [15,32–35,42,57,58]. A reduction in GMV was described in four articles [32,33,57,58]. A voxel-based morphometric (VBM) study reported a significant decrease in GMV within the bilateral prefrontal cortex in ME/CFS patients that correlated with pain severity [32]. Another study using the same method also showed a significant decrease in GMV in regions including the occipital lobes, right angular gyrus and the posterior division of the left parahippocampal gyrus in ME/CFS patients compared with HCs [58]. Two other studies linked the reduction of GMV in ME/CFS patients to a decrease in physical activity ability using VBM [36] and reduced GMV in the contralateral regions to ME/CFS symptom scores using longitudinal MRI [33]. The remaining two studies found no regional or global differences in GMV [35,42].

Nine papers mentioned significant WM changes in ME/CFS patients compared to HCs [11,14,15,31,33,34,42,58]. Four of the articles described a decrease in WM volume (WMV). A T1w and T2w MRI study found that WMV is decreased in the mid-brain of ME/CFS patients compared with HCs [11]. This finding was also consistent in two other studies using the same method [13,42], one which correlated this decrease with prolonged fatigue duration [13,42]. Another study that used phase-contrast, quantitative flow MRI reported significantly lower WMV in the pons and right temporal lobe in ME/CFS patients compared with HCs [34]. ME/CFS patients were also found to have significantly decreased WMV in the occipital lobes, the left inferior fronto-occipital fasciculus and adjacent areas [33,58]. Evidence of bilateral WM atrophy in ME/CFS patients has been reported and this was accompanied by an increase in cortical thickness in both arcuate end points, the middle temporal and precentral gyri, and the occipital lobe [31].

## Literature reporting changes in brain metabolites

Six studies investigated changes in brain metabolites this includes: N-acetylaspartate, choline and creatine [19,52,59,60,64,65]. Two studies reported an increase of choline- containing

compounds respective to creatine or unsuppressed water peaks [59,60]. One study found that higher temperatures in the right insula, putamen, frontal cortex, thalamus and cerebellum were associated with elevated lactate/ creatine ratios [65]

Ventricular lactate levels were assessed in three studies [19,52,64]. All studies reported significantly greater levels of ventricular lactate in ME/CFS patients compared to HCs [19,52,64]. One study associated this difference in ventricular lactate to an increase mental fatigue [52]. Another study reported a negative correlation between ventricular lactate levels and physical health and disability scores [19].

## Literature reporting changes in sleep quality

Sleep was investigated in three of the 48 studies. One of the studies found that ME/CFS patients had significantly reduced EEG alpha power, theta, sigma and beta spectral power during stage 2, slow wave sleep and rapid eye movement (REM) while delta power was reduced during slow wave sleep and then elevated during stage 1 and REM when compared to HCs [37]. Another study found that ME/CFS patients had lower occipital and central ultra-slow (US) delta power and higher occipital theta and alpha power compared with HCs. ME/CFS patients were also found to have significantly impaired subjective sleep quality compared with HCs [55]. A twin study found that there were no significant polysomnographic differences in REM latency, delta, fast frequency beta or alpha power between the ME/CFS twin and healthy twin [12].

An MRI study investigated structural differences that were associated with unrefreshing sleep in patient groups compared to HCs. Using the Pittsburgh Sleep Quality Index (PSQI), sleep quality of ME/CFS patients was significantly correlated with MRI signal intensities in the medial prefrontal cortex [42].

## Literature reporting changes in electrical activity

Nine articles discussed significant changes in energy waves using EEG [12,27,29,37,38,46–48,55]. These articles include the three previously mentioned papers that discussed EEG electrical activity in relation to sleep [12,37,55]. In addition to this, one article reported that increased ultra-slow (US) delta waves was approximately one fifth lower in ME/CFS groups compared with HCs. Theta, alpha, sigma and beta waves did not significantly differ between groups [38]. A twin study that used EEG Low-resolution electromagnetic tomography analysis (LORETA) found that there was significantly higher delta power in the left uncus and parahippocampal gyrus and higher theta power in the cingulate cortex and right superior frontal gyrus in the ME/CFS monozygotic twin compared with the healthy twin [27]. A study using the same method of analysis, LORETA also found hypoconnectivity in the delta, alpha and alpha-2 bands in patient groups compared to their respective HCs [29]. In another investigation small worldness, defined by the level of integration and clustering of neural networks, in the delta band was found to be significantly lower in ME/CFS patients compared with HCs [47]. Furthermore, one study detected lower beta 2 level current density in the somatomotor cortex, superior parietal lobe and medially in the precuneus and posterior cingulate of ME/CFS patients during a resting condition quantitative EEG (qEEG). The same study also found greater current density of delta frequency bands in ME/CFS patients compared with HCs [48]. A brain electrical mapping (BEAM) EEG study found that there were significantly elevated levels of delta, theta and alpha 1 waves in the right frontal and left occipital regions [46]. When requested to complete word finding and dot localisation cognitive tasks, significant differences in EEG source activity in the left frontal-temporal- parietal regions in ME/CFS patients were compared with HCs. Patients

were able to be successfully characterised at 83% by alpha band data when completing a cognitive task [39].

## Literature reporting changes in functional connectivity

Seven studies investigated changes in FC in ME/CFS patients compared with HCs [16,24,29,47,50,62]. In an ASL study, patients showed significant differences in FC relative to HCs. These differences include higher FC between regions including the bilateral superior frontal gyrus, anterior cingulate cortex (ACC), precuneus, and right angular gyrus to right postcentral gyrus, supplementary motor area, posterior cingulate gyrus and thalamus [16]. These differences strongly correlated with overall clinical fatigue. An ASL fMRI study showed both differences in static FC (sFC) and dynamic FC (dFC) [50]. This finding also linked disrupted FC with fatigue scores and has been corroborated by two other articles included in this review [24,62]. Another study using eLORETA EEG found hypoconnectivity in the delta, alpha and alpha-2 frequency bands in patient groups compared with HCs. Resting state connectivity deficits were also described in the occipital, parietal, posterior temporal and posterior cingulate in the ME/CFS cohort. A positive correlation between cognitive impairment and disrupted connectivity in the central executive network (CEN), salience network (SN) and default mode network (DMN) was also described in ME/CFS patients [29]. Another study using the same technique, found that there was significantly lower delta small-worldness in ME/CFS patients which was negatively correlated with neurocognitive impairment scores [47]. To support this, at resting state FC between the medial prefrontal cortex and inferior parietal lobules were weaker in the ME/CFS cohort, however, when performing a stroop task the FC was found to be significantly more complex [63].

## Literature reporting changes in cognitive function

There were 14 studies that reported on significant neurological findings in association with cognitive function in ME/CFS patients compared with HCs. A SPECT study reported that ME/CFS patient performance on the PASAT task was equivalent to HCs, however, activation of the anterior cingulate was much greater during the task [26]. In another study, when provided with a fatiguing task, there was no significant difference in responsiveness in task dependent regions in ME/CFS patients compared with HCs, however, patients had reduced response in the auditory cortices while controls had a constant response [45]. Using ASL reported higher fatigue levels in both ME/CFS patients and HCs using PASAT which was accompanied by increased CBF which then progressively decreased after three minutes. During recovery, when compared with HCs, ME/CFS patients exhibited a significant decrease in rCBF in the superior temporal gyri (STG), precuneus and fusiform gyrus [20]. Another study reported a significant increase in the activation of occipito-parietal cortex, posterior cingulate gyrus and para-hippocampal gyrus in ME/CFS patients compared with HCs following being subjected to a fatigue inducing task [23]. This same investigation reported a significant decrease in activation of dorsolateral and dorsomedial prefrontal cortex in ME/CFS patients relative to HCs. These findings were reversed in an anxiety-provoking situation [23]. A fMRI study using *n*-back tasks to measure working memory function found that the memory networks were activated, and performance levels were high in both ME/CFS patients and HCs. ME/CFS patients, however, had significantly reduced activation in the dorsolateral, prefrontal and parietal cortices and increased activation in the medial prefrontal regions including the anterior cingulate compared with HCs during 1-back condition. During the 2- and 3- back conditions ME/CFS patients also had significantly stronger large clusters in the right inferior/ medial temporal cortex. This activation was dependent on task load [22]. A BOLD study

found that ME/CFS patients had more extensive use of the working system network when provided with an audio processing task [40]. A later study saw that there were no differences in mental fatigue and brain activity in simple auditory monitoring tasks in both groups, however, there were marked differences in these in the cerebellar, temporal, cingulate and frontal regions in ME/CFS patients when requested to complete a working memory task [18]. This difference was evident following the control of baseline fatigue levels. Using the same technique, it was found that ME/CFS patients recruit a wider region of cortical and subcortical regions and had significantly lower SampEns characteristic of lower information capacity.

One EEG P300 study found that there was lower performance by ME/CFS patients in digit symbol and finger tapping task (FTT) compared with HCs [54]. There was no significant difference in performance in auditory verbal learning test (AVLT) nor digit span tasks.

In a rapid event-related fMRI, while similar neural networks were activated including the dorsal anterior cingulate cortex in both groups, the ventral anterior cingulate cortex was active in error trials in HCs, however, this was not activated in ME/CFS patients. Performance in ME/CFS patients was also significantly slower compared with HCs [57]. A BOLD fMRI study found that there was increased complexity in the posterior cingulate cortex in the default mode network in both resting state and during a stroop task. FC between the medial prefrontal cortex and inferior parietal lobules were significantly weaker in the resting state and more complex in the ME/CFS group during the task.

An fMRI study reported a significant decrease in activation of the right caudate and right globus pallidus in ME/CFS patients compared with HCs when subjects performed a gambling task. This correlated with an increase in mental fatigue, general fatigue and reduced activity [41].

When requested to perform a physical effort task, ME/CFS patients were found to have a reduced feedback related activity in the dorsolateral, prefrontal cortex proportional to state-related fatigue and prior beliefs about task performance ability compared to HCs.

## Literature reporting changes in secondary outcomes

Secondary outcomes were investigated in 43 out of 55 studies [11,13,14,16–33,35,36,38,41–48,50,51,54,57,58,62]. Secondary outcomes are presented in (S6 File). These outcomes included: quality of life (QoL), sleep, pain and fatigue scores. These were measured using a variety of different tools including Hospital Anxiety and Depression Scale (HADS), bell score, 36-Item short form survey (SF-36), multidimensional fatigue inventory (MFI), clinical interview schedule- revised (CIS-R) and visual analogue scale (VAS). Almost all studies found depression, anxiety, pain and fatigue scores were significantly higher in ME/CFS patients compared with HCs while physical function was higher in HCs. Cognitive impairment and mental fatigue were also measured and found to be greater in ME/CFS patients compared with their respective HCs in all cases where assessed.

## Quality assessment

To assess each article's quality and bias the JBI checklist was used (S7 File). The most commonly addressed item was item eight where 55 (100%) of the studies' outcomes were assessed in a standard, valid and reliable way [11–51,53–58,62]. Item six and seven followed where 49 (89.1%) of the studies were able to effectively identify confounding factors and mitigate them [11–45,47,48,50,51,53–55,57]. This was either achieved through inclusion or exclusion criteria, restrictions to certain classes of medications, caffeine or alcohol, and statistical adjustments. 48 (87.3%) studies addressed item one where they utilised methods to appropriately match ME/CFS patients and HCs [11–14,16–27,29,31,34–46,48,50,51,53–58,62]. This was most commonly achieved through age and sex-matching as well as weight and handedness- matching. Item two

was the least addressed checklist item by all included studies with only 20 out of 55 studies (36.4%) reporting source population information [12,18,21–24,26,27,29,30,32,34,37,41,45–47,53,62]. All twin studies were able to effectively adhere to this item through the condition that twins needed to be reared together [12,21,27]. While all papers included case-criteria definition used to select patients as assessed by item three, details on HC criteria were omitted or unclear in 11(20%) of the cases [17,28–30,32,36,47,49,55]. Item four, five and nine related to whether the exposure was measured in a standard and valid way, whether how it was measured was consistent across participants and HCs and if the exposure period was sufficient to show an effect. While these items were not applicable to all cases, the articles that did have an exposure were consistently administered across both case and control groups 100% of the time. The exposure period was sufficient to in all cases (100%) [17,18,20–23,25–28,30,34,40,41,43–45,50,57]. In most of the cases (20 out of 21), standard and valid measures for exposures was adhered to [12,17,18,20–23,25–28,30,34,40,41,43–45,50,57]. Thirty four articles (61.8%) included appropriate statistical analysis, this is inclusive of normalisation of datasets where necessary and adjusting for multiple comparisons [11,13,14,17–19,22–24,27–34,36,39,41–43,45,48,53,55,57,58].

## Discussion

Although there is no consensus on the pathomechanism of ME/CFS, neurological and cognitive dysfunction are key underlying features of this illness. The primary aim of this systematic review was to collect and appraise the literature related to the structural and functional neurological changes in ME/CFS patients as measured by neuroimaging techniques.

The mean age across all studies was 40.9 years for ME/CFS patients and 39.1 years for HCs. When considering the average illness duration of patients which was 9.68 years, this is consistent with epidemiological data that reports that onset of ME/CFS predominantly occurs around 29–35 years [66]. A greater proportion of participants included in this analysis were female (3:1, female: male), this was due to the preponderance of women who have ME/CFS [67]. Some studies [12,16,24,35,36,38,39,50,51,54,57,62] only selected females due to availability and to control for morphological and physiological differences in the brain and associated structures between sexes when pooling data [62]. These include differences in GM density and volume. For studies that did not appropriately sex-match their participants, this was a major shortcoming of their study as they could not account or adjust for these significant differences between sexes [62].

From the 55 studies, 14 handedness-matched the participants, this involves grouping patients on the basis of being right-handed, left-handed or ambidextrous. This was a conservative approach to account for potential atypical lateralisation and other implications handedness has on brain structure and function. In a VBM study of 465 healthy adult brains they found minimal effect of handedness on symmetry [68]. This is supported by a review article that assessed the effect of handedness on the findings from numerous different neuroimaging techniques [69]. Therefore, handedness matching may not be a necessary consideration for future imaging studies.

In 14 studies, participants were weight or BMI matched. This was an effective measure to reduce the impact weight differences may have on neurological morphology and function; including, GMV and functional activation and/ or connectivity. As physical exercise also contributes to these differences, three studies (6.25%) recruited only sedentary controls [18,21,53]. Recruitment of only sedentary HCs to control for the effect of exercise on brain morphology and function is an important consideration for future studies. Three of the included papers were monozygotic twin studies [12,21,27]. Each twin pair shared at least 99.9% zygosity and

comprised of a healthy twin and one discordant for ME/CFS [12,21,27]. The inclusion of an unaffected twin as a HC is effective in limiting the impact of genetic differences on brain morphology [12,21,27,70]. As the included studies had twins that were reared together, this might also effectively control for differences in environmental exposures [12,21,27]. There may be a limitation to using twin studies as ME/CFS has a significant genetic component and the effect this may have on the twin without a current diagnosis of ME/CFS is not known [71].

Out of the 55 papers, 43 relied on the Fukuda criteria to classify ME/CFS patients. With the requirement of only four key symptoms for a diagnosis, this criterion has been criticised for being too broad [6]. In addition to this, there is considerable overlap of key symptoms with other non- ME/CFS illnesses. Due to this potential overlap, the inclusion of additional patient characteristic information such as infection onset is also an important consideration for future studies. CCC and ICC-defined patients are a more homogenous subset of patients [7,8]. Across these studies there were only a small proportion of patients that fell under CCC and no patients included that were defined by ICC or IOM criteria [11,13,15,20,29,33,42,62]. This may be due to the lack of availability of participants that fall under these more refined CCC, ICC or IOM criteria. As Fukuda patients are a more heterogeneous subset of patients, this may explain the variability seen both within and between the studies. Inclusion of the Fukuda criteria was a limitation of our search strategy, however, due to this criteria being still abundantly used in research and clinical practice, limiting our search to not include papers that relied on the Fukuda criteria would drastically reduce the availability of appropriate papers in this field [72].

All these studies use standard clinical procedures to assess different aspects of neurocognitive function. While magnetoencephalography articles were searched, based on inclusion and exclusion criteria none of them were included in this review. Across all studies, no two used the exact same method or came to the same findings.

With the studies that investigated CBF in ME/CFS patients relative to controls there were a variety of results documented. This includes higher CBF volume reported in patients as well as no significant differences when compared to HCs. CBF values are sensitive to many parameters including exercise, cardiovascular health and blood pressure. OI is a symptom observed in a subset of ME/CFS and this also has been found to influence CBF levels [53,73]. However, it should be noted that the presence of OI was not consistently documented in these studies. Based on these results it is evident that CBF is an inconsistent correlate of ME/CFS.

The available literature shows conflicting evidence regarding changes in GMV in ME/CFS patients compared with HCs. A study conducted by de Lange *et al* reported lower levels of GM in ME/CFS patients [34,36]. These findings, however, were unable to be replicated when conducted in a larger representative sample [34–36]. This follow-up study by Van der Schaaf *et al* [34] instead found no regional change in GMV in ME/CFS patients compared with HCs, a finding also corroborated by Shan *et al* and Barnden *et al.* [74]. In contrast, Finkelmeyer *et al* reported an increase in GMV in ME/CFS females compared with HCs. As with rCBF, GMV changes are sensitive to a plethora of interferential factors including chronic pain, stress, age, physical activity and insomnia [74]. As these features are variable, it is difficult to control for them and draw appropriate conclusions on GM volumes in ME/CFS patients.

WM changes including WMV [11, 13,14,15] and WM atrophy [31] were reported [11,14,15,31,33,34,42,58]. Bilateral WM atrophy accompanied by increased cortical thickness in both arcuate end points, the middle temporal and precentral gyri and the occipital lobe was described by Zeineh *et al* [31]. Four studies reported a decrease in WMV in ME/CFS patients in regions including the mid-brain, pons, right temporal lobe, occipital lobes and left IFOF [13,33,34,58]. Depression and anxiety often co-exist with ME/CFS and these disorders have been found to have disrupted WM architecture. A study conducted by Barnden *et al* [13] proposed that MRI differences in ME/CFS exist independent of the influence of these conditions.

This article predicts that impaired brainstem nerve conduction is a result of depleted brainstem myelin. Impaired brainstem conduction can consequently lead to disrupted sensory, motor and cognitive function [75] and could explain multiple symptoms observed in ME/CFS patients including autonomic dysfunction, impaired cognition and sleep disturbance [11,13]. A more recent article by Barnden et al. [15] supported this case and suggested that this may be due to an upregulation of myelin in the sensorimotor cortex which was a potential compensatory mechanism to maintain brainstem-sensorimotor connectivity.

Inflammation of the midbrain along with cingulate, thalamus, amygdala, pons and hippocampus was reported by Nakatomi et al. [25]. While the underlying cause of the neuroinflammation in ME/CFS is unknown, overactivity of neurons resulting in the production of proinflammatory cytokines, reactive oxygen species and nitrogen species was proposed as a potential mechanism [25]. Shungu et al. [19] found that ventricular lactate and cortical GSH were inversely correlated. As they detected an increase of ventricular lactate in ME/CFS patients, depleted GSH was also observed. It was suggested that this was an indicator of redox dysregulation and increased oxidative stress [19]. This finding was supported by two other studies [52,64]. Studies that investigated this may be potentially due to Miller et al. [41] who described neuroinflammation as a potential mechanism underlying impaired neurotransmission in the basal ganglia. These features strongly suggest a neuroimmune process underlying ME/CFS pathology.

The most common morphological changes occurred within the cortical regions. The cortical regions are important for higher order processing and includes the cingulate cortex [76]. The cingulate region has broad roles in emotional, cognitive and error processing [77]. Changes in the cingulate cortex have also been associated with cognitive disruption, and this was observed in three of the included studies. Disruptions to working memory was also a consistent feature shown across studies that investigated cognitive ability in ME/CFS patients compared with HCs.

The cingulate cortex has strong reciprocal connections to the orbitofrontal cortex, the basal ganglia and the insula. The basal ganglia have important roles in reward-processing. Dysfunction of the basal ganglia has also been implicated in ME/CFS [41]. A fMRI study by Miller et al. [41] found that when performing a monetary gambling task, the right caudate and right globus pallidus were found to have significantly less activation in ME/CFS patient groups compared with HCs. The decreased activation detected in the right globus pallidus correlated with increased mental fatigue, general fatigue and reduced activity [41].

Disruption of the hippocampus was also observed in six studies. In the study conducted by Nakatomi et al. [25] they found that the binding potential of 1C-(R)-PK11195, a marker for neuroinflammation, in the hippocampus correlated with depression scores in ME/CFS patients. Significantly reduced serotonin receptors were also detected in ME/CFS particularly in the hippocampus. Depression is a comorbidity prevalently seen in patient groups as shown in the secondary measures and has been strongly associated with changes in this structure [78]. While the role of the hippocampus in ME/CFS pathology is not fully understood this indicates the importance of controlling for these comorbidities either statistically or through more stringent criteria.

Due to having strict inclusion and exclusion criteria on the basis of Cochrane guidelines, one paper, Kimura et al (2019) that was within the scope of the review, was excluded on the basis of not containing two key words in the abstract and title. This paper identified microstructural changes in ME/CFS using the MRI technique diffusional kurtosis imaging and neurite orientation dispersion and density imaging. A feature found in ME/CFS patients compared to HCs was that there was significantly lower mean kurtosis in the right superior longitudinal fasciculus. This pathway is involved in processes such as emotion, language, attention and

memory and could be altered in ME/CFS patients, the study, however, is limited by sample size and the contribution of potential gender bias [79].

Disrupted FC was consistently reported in all six papers that conducted a FC analysis. The central executive CEN, SN and DMN are the most referred to intrinsic connectivity network relating to ME/CFS pathology. These networks are highly interconnected and disruptions to all three networks were correlated to cognitive dysfunction in ME/CFS patients. Abnormalities in the absence of a task were reported for all three and can provide insight into performance deficits with cognitive tasks [62].

Sleep disturbances have been described in ME/CFS patients, this is inclusive of: changes in sleep efficiency, unrefreshing sleep and sleep fragmentation [37,55]. A common trend found within these studies is that objective sleep measures did not correlate with subjective reports provided by ME/CFS patients. It was proposed that this difference may have arisen from a misperception. Shan *et al.* [42] found that there were brain structural differences in the medial prefrontal cortex that were associated with unrefreshing sleep, suggesting that there are structural impairments related to the subjective measure of impaired sleep quality. There were numerous more sleep studies investigating ME/CFS, however these are not covered in the scope of this review. Sleep specific—phenotypes and their association with ME/CFS patients still has not been determined.

Multiple differences in EEG activity have been reported. Decreased ultra-slow delta power in ME/CFS patients has also been reported [38]. These US delta power results suggest that there may be impairment at the level of cell membrane potential oscillations or neural recruitment. This shows the importance of going beyond conventional EEG bands [38]. BEAM identified significant differences in delta, theta and alpha 1 in the right frontal and left occipital regions between ME/CFS patients and HCs. This all suggests reduced activity in ME/CFS patients. The findings also indicate a decrease in overall complexity suggestive of relatively inhibitory state in brain function [46].

Due to functional impairment observed in the autonomic nervous system (ANS) of ME/CFS patients, the system may undergo progressive changes to adjust and compensate for this impairment. Numerous potential examples of ANS compensatory mechanisms were present across the studies; this includes, midbrain changes in nerve conduction resulting in downstream adaptation of the same circuits and upregulation of myelin in ME/CFS patients compared with HCs. Increases in FC in patient groups in response to disrupted FC elsewhere is another possible example of compensation [50,80]. Shan *et al* [33] also described neural plasticity through a longitudinal MRI study by assessing changes in left IFOF WM volumes over six years in patients compared to respective controls. While the effect of plasticity on the nervous system and ME/CFS is not well understood, longitudinal studies may be an important consideration for future studies in contrast to cross-sectional studies to better describe ME/CFS specific changes.

The search for definitive cerebral signatures for ME/CFS remains difficult. The lack of reproducibility of the studies and low sample size also contribute to the inability to arrive at a consensus regarding potential neurological markers for ME/CFS. There are, however, consistent examples of altered brain health, e.g. an increase in neuroinflammation suggestive of a neuroimmune mechanism and reduced functional efficiency.

## Quality assessment

Quality levels were variable across the included studies in this review. All studies utilised standard measures for clinical evaluation of neurological changes including MRI, fMRI, PET and EEG. While all papers included selection criteria for ME/CFS patients, criteria for HCs were

often omitted or not considered. Studies that applied two or more forms of patient-control matching, of which one had to be sex-matching, were deemed as successfully addressing item 1: whether the groups were comparable other than the presence of disease in cases or the absence of disease in HCs. This is due to the structural and physiological neurological differences between males and females, as well as discrepancies caused by weight and age. A large proportion of the studies identified potential confounding variables and control measures; this includes sex-, weight and age- matching, exclusion of certain medication classes, restriction of caffeine and alcohol as well as statistical adjustments for confounding factors such as anxiety and depression. The least addressed criterion was item two. This criterion assesses whether patient and HC groups were appropriately matched with regards to socio-demographic characteristics. This level of detail was only included in 39.6% of the studies in this review. One limitation of this quality assessment is that while quality measurement is on a spectrum level, the responses to these checklist questions provided could only be addressed at a categorical level, thus possibly introducing inter-reviewer bias. Recommendations for future studies in this field include independently analysing data from both sexes, referencing patient socio-demographics, providing additional information including comorbidities and medications.

## Future directions

Validation of these findings reported in this review with larger sample sizes is necessary. Since the publication of many of these articles, the Fukuda definition is used less prominently in clinical and research practice. Emerging research will utilise the CCC, ICC and the IOM criteria more frequently. This change will allow for standardise patient cohorts to be analysed. Additionally, advancing technologies will also improve this area of research including the introduction of 7T MRI which offers improved spatial and contrast resolution; this may assist in discerning these findings. Use of machine learning tools such as ensembles will also revolutionise this field particularly in improving robustness, reducing error and allow recognition of imaging signatures at a single individual level [81].

## Conclusion

This is the first systematic review that collected and appraised available literature relating to these neurological changes in ME/CFS patients as measured by neuroimaging techniques. Although there is a lack of agreement on the origins of this illness, ME/CFS patients exhibit widespread autonomic disruption inclusive of morphological changes, WM abnormalities and aberrations in FC. These characteristics were, however, not consistent across the studies and further research is required to understand neurological involvement in ME/CFS pathology.

## Supporting information

**S1 File. PRISMA 2009 checklist.**
(DOC)

**S2 File. Extended search term code.**
(DOCX)

**S3 File. Summary of study characteristics table.**
(DOCX)

**S4 File. Summary of participant characteristics table.**
(DOCX)

**S5 File. Summary of primary outcome results table.**
(DOCX)

**S6 File. Secondary outcome measures table.**
(DOCX)

**S7 File. JBI quality assessment table and descriptions.**
(DOCX)

## Author Contributions

**Conceptualization:** Leighton Barnden, Donald Staines, Sonya Marshall-Gradisnik.

**Data curation:** Rebekah Maksoud, Stanley du Preez.

**Formal analysis:** Kiran Thapaliya.

**Methodology:** Rebekah Maksoud, Stanley du Preez, Natalie Eaton-Fitch.

**Supervision:** Hélène Cabanas, Donald Staines, Sonya Marshall-Gradisnik.

**Writing – original draft:** Rebekah Maksoud.

**Writing – review & editing:** Rebekah Maksoud, Stanley du Preez, Natalie Eaton-Fitch, Kiran Thapaliya, Leighton Barnden, Hélène Cabanas, Donald Staines, Sonya Marshall-Gradisnik.

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
