## [Decision Letter · Decision Letter 0]

21 Feb 2020

PONE-D-19-34471

A Systematic Review of neurological and cognitive impairments in myalgic encephalomyelitis/ chronic fatigue syndrome using neuroimaging techniques.

PLOS ONE

Dear Miss Maksoud,

Thank you for submitting your manuscript to PLOS ONE. After careful consideration, we feel that it has merit but does not fully meet PLOS ONE’s publication criteria as it currently stands. Therefore, we invite you to submit a revised version of the manuscript that addresses the points raised during the review process.

Reviewers and I agreed major revision is required. I would especially recommend you improve search criteria/paper inclusivity, and better discuss study findings. Please, find below detailed comments.

We would appreciate receiving your revised manuscript by Apr 06 2020 11:59PM. To enhance the reproducibility of your results, we recommend that if applicable you deposit your laboratory protocols in protocols.io, where a protocol can be assigned its own identifier (DOI) such that it can be cited independently in the future. For instructions see: http://journals.plos.org/plosone/s/submission-guidelines#loc-laboratory-protocols

We look forward to receiving your revised manuscript.

Kind regards,

Marcello Moccia

Academic Editor

PLOS ONE

Additional Editor Comments (if provided):

Thank you for submitting your interesting manuscript. Reviewers and I agreed major revision is required. I would especially recommend you improve search criteria/paper inclusivity, and better discuss study findings. Please, find below detailed comments.

Journal Requirements:

2. Please include a copy of Table 5 which you refer to in your text on page 15.

Reviewers' comments:

Reviewer's Responses to Questions

**Comments to the Author**

1. Is the manuscript technically sound, and do the data support the conclusions?

Reviewer #1: Partly

Reviewer #2: Partly

Reviewer #3: Yes

2. Has the statistical analysis been performed appropriately and rigorously? 

Reviewer #1: N/A

Reviewer #2: N/A

Reviewer #3: Yes

3. Have the authors made all data underlying the findings in their manuscript fully available?

Reviewer #1: Yes

Reviewer #2: Yes

Reviewer #3: Yes

4. Is the manuscript presented in an intelligible fashion and written in standard English?

Reviewer #1: Yes

Reviewer #2: Yes

Reviewer #3: Yes

5. Review Comments to the Author

Reviewer #1: This is a systematic review on ME/CFS research using neuroimaging techniques. The authors selected 48 articles based on procedures commonly applied in systemic reviews.The authors picked up various imaging techniques including MRI, SPECT and EEG, which provide precious information on this topic.In total, the reviewer thinks it is an important work, which potentially indicate future directions.However, the reviewer think there are some arguments which are shown below.

1.The review process is not completely clear.

The authors finally picked up 48 papers and discussed in detail.Although the authors described the details of the process (in Fig.1), the reviewer are not completely convinced with the final selection process, where 98 articles were excluded. Below are examples the reviewer does not understand why they are not included.

Munemoto, T., Soejima, Y., Masuda, A., Nakabeppu, Y., & Tei, C. (2017). Increase in the regional cerebral blood flow following Waon therapy in patients with chronic fatigue syndrome: a pilot study. Internal Medicine, 56(14), 1817-1824.

Kimura, Y., Sato, N., Ota, M., Shigemoto, Y., Morimoto, E., Enokizono, M., ... & Sato, W. (2019). Brain abnormalities in myalgic encephalomyelitis/chronic fatigue syndrome: Evaluation by diffusional kurtosis imaging and neurite orientation dispersion and density imaging. Journal of Magnetic Resonance Imaging, 49(3), 818-824.

Both papers are from Asia (Japan). A paper by Munemoto et al utilize SPECT and a paper by Kimura et al describe MRI abnormalities of ME/CFS patients as compared to healthy subjests using diffusion kurtosis imaging.Such study using new techniques would shed light on new aspects of ME/CFS.

2. Comparison between ME/CFS and disease-controls.

Although the paper focus on the difference between ME/CFS and healthy subjects, the reader would be interested in the comparison with other diseases as well. How about mentioning about the studies (or reviews) comparing ME/CFS and other diseases (for example,comparisons between ME/CFS and depression with regards to SPECT et ctc.). Even if there are no good studies ever performed, the authors could discuss this point in the discussion section (as a future direction).

3. Description about the 'normal-appearing' or 'non-specific' findings.

It is known that in conventional MRI, ME/CFS brain looks basically normal(different from neuroimmunological diseases such as multiple sclerosis). The authors are advised to discuss on this point. It was pointed out that there exists some 'non-specific' T2 high spots scattering in the ME/CFS brain. What does it mean? Is there any new analysis on this point? The review would be improved by including such information.

4. Future directions.

In the discussion section, it would be better to include 'future direction' section, which might include a potential advantages of new technologies such as 7T MRI.

Reviewer #2: Peer review: PLOS ONE

Neurological and cognitive impairments

This is a most welcome effort that can help the ME/CFS field to better organize and understand the neurological component of symptoms, and we appreciate the care and work that must have gone into this draft. Comments and critiques are given in the spirit of making a potentially worthy resource even better.

Our primary critiques and suggestions center around organization, inclusivity of papers, and descriptions of research findings.

Organization: in general, the manuscript would greatly benefit from editing and organization.

The Quality Assessment was described twice (the second description was more concise and clear than the first). The “Discussion” section reads more like a separate draft of the earlier findings, as opposed to a gestalt summary of them. It is somewhat unclear what the overall organization scheme is - for example “Literature reporting changes in brain structure” and “Literature reporting changes in gray and white matter” are currently separate sections, but these two sentences are synonyms: structural studies are, by definition, studies of gray and white matter. What is currently “changes in brain structure” now includes both structural and functional studies, which are traditionally considered separate categories of dependent measures in neuroimaging. The paper should be read carefully, front to back, by someone with neuroimaging expertise so these kinds of mistakes can be fixed. Another example is Line 187 in which reference 57 is listed among studies of hippocampal function, but it is a study of hippocampal structure and not function. Line 411 begins the header “Neurological and cognitive changes…” but then grey matter volume and white matter volume studies follow.

The paper should be carefully edited by someone with neuroimaging expertise so these kinds of errors can be fixed. Scattered throughout the paper are other small errors, for example Line 270 describing arterial spin labeling as a measure of functional connectivity (it is not), and what appears to be a conflation of hippocampus and hypothalamus (Line 482). A careful reading by a neuroimaging expert should catch these small errors - Given the stigma surrounding this illness, it is imperative that published work portrays expertise and accuracy, the current draft does not.

It is also important that patients who will read this article do not mistakenly believe that each reported finding represents an established biomarker. A neuroimaging review is a useful potential resource, but in general the current draft lacks contextualization. There are example of contradictory findings, for example starting at Line 167, the section on cingulate - this is a very large and heterogenous brain region, and the reported findings are sometimes directly contradictory (e.g., separate findings describing both hyperactivation and hypoperfusion in posterior cingulate/ praecuneus). These inconsistencies should be pointed out, currently they are glossed over in a list of findings.

There are some things that are unnecessarily stated multiple times (e.g., the single report of fractional anisotropy in arcuate fasciculus, lines 208, 222, 231, 434, 475, and Table 3). This inappropriately gives the impression of a definitive and well-founded biomarker finding when in fact it was a small (N=15 patients vs 14 controls) and unreplicated study. This sort of contextualization of findings is important for a review article; one can picture a patient inappropriately telling his/her physician that they have an arcuate fasciculus problem. The reality is that most of the reviewed studies are small and unreplicated, and some (if not most) findings are likely to represent Type 1 errors in a heterogenous diagnosis - this should be made more clear in an Introduction or a Discussion section. General principles such as loss of volume due to poor brain health, increased infection/neuroinflammation-associated markers, and reduced functional efficiency could be described, so that the paper is more than a laundry list of disparate reports.

The current draft reads like a review of neuroimaging studies - a very valuable resource - but not a review cognitive studies. Given that the focus is on neuroimaging per se and not cognitive dysfunction, “Cognitive” should probably not be in the title: for example, Gotts et al (2015) PLoS ONE is a study of sleep and cognition - two topics that are covered in this review. But it is not referenced or discussed in the current review because it was not a neuroimaging study. “A systematic review of neuroimaging….” may be an example of better phrasing for the title. If there is to be a focus on cognition per se, even outside of the scanner, then there should be an approach more like Cvejic et al. (2016), including search terms related to cognitive testing.

The literature search may benefit from some more inclusive terms, and from the use of other review articles.

Line 65: “Myalgic encephalomyelitis” should be listed among the search terms, as should MECFS, ME/CFS, CFSME, and other iterations. “Magnetic resonance spectroscopy” and related iterations should also be included.

The paper 'Costa et al (1995) Brainstem perfusion...' appears to have been missed; it used “SPET” single photon emission tomography, this may have been missed by using the more specific and constrained search of “tomography, emission-computed, single-photon” - perhaps simply using “tomography” would be more inclusive and it would also catch PET studies.

We suggest that the authors include “magnetic resonance spectroscopy” in the search terms to make the study list more inclusive. For example, Murrough et al. (2010), NMR in Biomedicine, Mathew et al. (2008), NMR in Biomedicine. From the 10 spectroscopy papers listed in VanElzakker, Brumfield, & Mejia 2019, only 2 were included in the current review. There have been a handful of neuroimaging reviews that could also be a source for appropriate papers, by looking through references:

Morris, Berk & Puri (2018); VanElzakker, Brumfield, & Mejia 2019;

www.meresearch.org.uk/information/publications/structural-and-functional-neuroimaging

See also Byron Hyde’s books.

We were surprised that a vast majority of studies were based on Fukuda criteria. One key problem with the Fukuda definition is that it likely includes individuals who deal with fatigue but do not share the same pathophysiology as ‘true’ ME/CFS. Inclusion of “Infectious Onset” as a ME/CFS cohort descriptor would be particularly helpful, if this is not possible it should be described as a limitation.

Line 68: Please correct this numbering for the supplementary material (listed as S1 but is actually S2). This problem occurs throughout the paper. Please address all mentions of supplementary materials.

Line 85: IOM criteria not listed here; given its prominence, if this was deliberate it should be stated explicitly. This is especially salient given the preponderance of Fukuda papers referenced.

Line 95

The authors describe that this manuscript intended to review and survey studies comparing ME/CFS to healthy controls, but it appears that some papers which make such a comparison were not included simply because they also made other comparisons. There are some examples of papers that have more than two groups - for example, Natelson et al (2017), Fatigue Biomedicine Health & Behavior 5(1). Given that there was a comparison between ME/CFS and healthy controls in that paper, it should probably be included (and the comparison to fibromyalgia can be ignored, in accordance with the self-imposed criteria for the current review). Other examples of this should be looked for.

Line 96

Please explain the reasoning for excluding sleep intervention studies. Was it all intervention studies that appeared in your search that you excluded or solely the sleep intervention studies? Please explain if this distinction was made and why. Was the exclusion only for studies that attempted to alter or influence sleep?

Line 113: the PubMed search term list was unusually short compared to the others, please explain.

Line 128, did any studies report race or ethnicity?

Line 136, in general, sentences should not begin with numerals (see elsewhere as well, e.g., Line 121) - a minor formatting issue throughout that should be edited. Line 328 vs line 329, “Table” and “Figure” should have consistent capitalization throughout.

Line 139: we believe the authors meant “spectroscopy” - this is technically an fMRI technique (functional). In general, neuroimaging studies are usually described according to a distinction between functional and structural; this distinction could be made clear throughout the paper.

Line 147 describes MRS and finding a difference in rCBF, but this is not something that MRS can do. We believe the authors intended to describe the ASL technique’s finding. In general, there were several instances in which more than one technique was used (MRI machines can of course do multiple methods of structure and function) and a result was described according to a different method. ASL is usually considered a form of fMRI (it describes function, not structure).

Line 163: Did the authors intend to write “intercranial compliance” ? In general, this is an example of a finding that would benefit from even a brief interpretation or explanation.

Line 187 - (organisation) hippocampus function is listed in a section on brain structure, and EEG findings are not listed in the section on electrical activity.

Line 189 - no comma after “Significantly”

Line 200 - It is true that myelin levels may be associated with T1 signal; however, studies examining myelin usually use a T1/T2 ratio and not simply T1 signal, which is much harder to interpret. It should be made clear that “myelin levels” is an interpretation of a finding as opposed to an empirical finding (elsewhere in the paper as well). Relatedly, Line 203 should explicitly state what variables were correlated, and take care to describe the difference between the reported statistical finding and its proposed interpretation.

Line 207 - the word “study” should follow “DTI”, line 208 the term FA appears unnecessarily; the word “arcuate” is misspelled

Line 213-220 - terms like “decrease” used as a verb sounds like longitudinal study findings; this wording should be clarified in case-control comparisons that cannot delineate a pre-existing risk factor from a disease process (i.e., smaller volumes may be the cause or may be the effect of a disease)

Line 257 - Another example of a reported finding that would benefit from even a small amount of explanation, example wording could be something like “small worldness, or propensity for local functional connectivity over long-range functional connectivity...” Line 264, Citation 46 - this is another example of why contextualization is important: the “inhibitory state” was an interpretation of results in a study comparing 24 patients to 23 controls, and was not an empirical finding. Again, the sleep & electrical sections could benefit from a small overall summary so that it is not simply a list of papers.

Line 291 - patients had or patients exhibited; not patients reported

Line 298 - typos

Line 302 unclear wording, line 309 unclear wording; but more importantly: line 302 vs line 308 different auditory tasks appear to have a positive vs a null finding. Are the two studies comparable? Are they inconsistent? Is there a possible conceptual or mechanistic reason for why these two studies seemed to find different things? An example where contextualization would be helpful.

Line 312 - dACC is usually not referred to as a nucleus

Line 314 - a finding that needs some explanation or contextualization

Line 333 - briefly define ‘role function’ - also wording in 353 - 356, explain ‘exposure period was sufficient to’

Line 337 - Organisation: Quality assessment seems more like a “Methods” topic, used in the initial selection of papers. In general, one cannot read this section and understand what the authors did - if these methods are truly central to paper quality assessment, this table should either be moved from supplemental into the main paper, or the text should be much more descriptive and not rely upon flipping back and forth between the text and the table to understand what is being described. Line 523 begins a clearer description of what seems to be described in 337.

Line 362 - begins the “Discussion” but what follows seems more like main text, including descriptions of the participants & study characteristics. Line 411 is a section titled “Neurological and cognitive changes in ME/CFS patients” which appears to be very similar to the sections in the earlier half of the paper. The “Discussion” section should probably be relatively short and offer some take-away messages about the overall findings. It should probably not introduce new topics, except to briefly point out limitations or future directions.

Line 431, references should follow each statement separately - in other words, studies that reported white matter hyperintensities should follow “WM hyperintensities” - white matter atrophy and white matter volume are synonymous, but references supporting this statement should follow that statement specifically.

Line 432 - “not of pathological significance” is an interpretation. Line 438 “established” is too strong; the author is making an argument based upon interpretation of data (this also directly contradicts line 482); 444 - “upregulation of myelin” is an interpretation and not an empirical finding.

Line 486 - define ICN, and please make sure all other acronyms are defined

Line 487 - This sentence fragment seems like evidence that the paper was not proofread by all authors. Please assure that all authors have read and edited the paper before resubmission.

Line 492 - hippocampus is itself a limbic structure. Was 'hippocampus' and 'hypothalamus' again switched? One does not normally think of hippocampus as central to arousal.

Line 518 - “compensation” is an example of an interpretation or theory, not an empirical finding, and should be framed as such.

Table:

For the table, the term USA is more appropriate, because Brazil is in “America” but this does not seem like important information when judging the quality and findings of studies. Perhaps a “Method of analysis” column has room for more information, for example it could include the Tesla strength of MRI. The “sample type” column did not seem important given that all entries said “CFS/ME.” The table also lists all studies as “prospective” but we believe this may be a misapplication of the term, which would mean that subjects were scanned before they developed the disease (see also Line 122, 123, 124). http://sphweb.bumc.bu.edu/otlt/MPH-Modules/EP/EP713_AnalyticOverview/EP713_AnalyticOverview3.html

It is again important that senior authors carefully read the entire paper.

Reviewer #3: The article is a systematic review of the various neurological and cognitive defects in ME/CFS. There has been extensive research investigating neuroinflammation and associated impairments in ME/CFS, using various imaging modalities. The authors have done a good job in compiling the various pertinent aspects. This paper will be of interest to existing researchers in the field, as well as for new researchers.

There are a few issues that the authors needs to address:

1. Why was Fukuda the primary chosen criteria? As the authors have rightly pointed out, it is too broad according to the majority of researchers and clinicians.

2. Are there no articles available using some of the more restrictive criteria (IOM for example?) that the authors could have used?

6. PLOS authors have the option to publish the peer review history of their article (what does this mean?). If published, this will include your full peer review and any attached files.

Reviewer #1: No

Reviewer #2: No

Reviewer #3: No

---

## [Author Response · Author response to Decision Letter 0]

31 Mar 2020

Reviewer #1: 

Reviewer #1, comment 1: The review process is not completely clear. The authors finally picked up 48 papers and discussed in detail. Although the authors described the details of the process (in Fig.1), the reviewer are not completely convinced with the final selection process, where 98 articles were excluded. Below are examples the reviewer does not understand why they are not included.

Munemoto, T., Soejima, Y., Masuda, A., Nakabeppu, Y., & Tei, C. (2017). Increase in the regional cerebral blood flow following Waon therapy in patients with chronic fatigue syndrome: a pilot study. Internal Medicine, 56(14), 1817-1824.

Kimura, Y., Sato, N., Ota, M., Shigemoto, Y., Morimoto, E., Enokizono, M., ... & Sato, W. (2019). Brain abnormalities in myalgic encephalomyelitis/chronic fatigue syndrome: Evaluation by diffusional kurtosis imaging and neurite orientation dispersion and density imaging. Journal of Magnetic Resonance Imaging, 49(3), 818-824.

Both papers are from Asia (Japan). A paper by Munemoto et al utilize SPECT and a paper by Kimura et al describe MRI abnormalities of ME/CFS patients as compared to healthy subjests using diffusion kurtosis imaging.Such study using new techniques would shed light on new aspects of ME/CFS.

Authors Response: Considered and unmodified in the manuscript. Munemoto et al included a participant that was under 18 and used an alternative paediatric case definition that was not covered in our inclusion/ exclusion criteria. Kimura et al, was excluded during the abstract/ title screening process as it did not contain any of the listed key words. 

Reviewer #1, comment 2: Comparison between ME/CFS and disease-controls. Although the paper focus on the difference between ME/CFS and healthy subjects, the reader would be interested in the comparison with other diseases as well. How about mentioning about the studies (or reviews) comparing ME/CFS and other diseases (for example,comparisons between ME/CFS and depression with regards to SPECT et ctc.). Even if there are no good studies ever performed, the authors could discuss this point in the discussion section (as a future direction).

Authors Response: Considered and unmodified in the manuscript. Reviewing studies on the neurological differences between ME/CFS patients and diseased controls is a detailed process that justifies its own investigation and was beyond the intended focus of this review. 

Reviewer #1, comment 3: Description about the 'normal-appearing' or 'non-specific' findings. It is known that in conventional MRI, ME/CFS brain looks basically normal(different from neuroimmunological diseases such as multiple sclerosis). The authors are advised to discuss on this point. It was pointed out that there exists some 'non-specific' T2 high spots scattering in the ME/CFS brain. What does it mean? Is there any new analysis on this point? The review would be improved by including such information.

Authors Response: Considered and modified in the manuscript. Descriptions on the results from all the included studies have been provided. The statement “The search for definitive cerebral signatures for ME/CFS remains difficult” (line 524) has been included to emphasise that there are not consistent cerebral markers found yet. 

Reviewer #1, comment 4: Future directions. In the discussion section, it would be better to include 'future direction' section, which might include a potential advantages of new technologies such as 7T MRI.

Authors Response: Considered and modified in the manuscript. A future directions section has been added to the discussion. This section reads as: validation of these findings reported in this review with larger sample sizes is necessary. Since the publication of many of these articles the Fukuda definition is used less prominently in clinical and research practice. Emerging research will utilise the CCC, ICC and the IOM criteria more frequently. This change will allow for standardise patient cohorts to be analysed. Additionally, advancing technologies will also improve this area of research including the introduction of 7T MRI which offers improved spatial and contrast resolution; this may assist in discerning these findings. Use of machine learning tools such as ensembles will also revolutionise this field particularly in improving robustness, reducing error and allow recognition of imaging signatures at a single individual level (line 548). 

Reviewer #2: 

Reviewer #2, comment 1: The Quality Assessment was described twice (the second description was more concise and clear than the first).

Authors Response: Considered and modified in the manuscript. The following details have been added to the quality assessment (QA) methodology: Each checklist item assesses the following: (1) group matching, (2) source population, (3) criteria, (4) method of exposure, (5) assessment of exposure, (6) identification of confounding variables, (7) management of confounding variables, (8) measurement of outcomes, (9) exposure period selection, (10) statistical analysis (line 111). Details of the methodology has also been removed from the discussion section. Upon consultation of guidelines and correspondence with a PLOS ONE editorial team member (31st March correspondence) the quality assessment table is required to be presented as supporting information and can’t be moved to the main text. 

Reviewer #2, comment 2: “Literature reporting changes in brain structure” and “Literature reporting changes in gray and white matter” are currently separate sections, but these two sentences are synonyms: structural studies are, by definition, studies of gray and white matter. What is currently “changes in brain structure” now includes both structural and functional studies, which are traditionally considered separate categories of dependent measures in neuroimaging.

Authors Response: Considered and modified in the manuscript. The subheading has now been changed to say “literature reporting changes in brain structure and function” (line 167). Literature reporting changes in gray and white matter has now been moved to the changes in brain structure and function section (line 211). 

Reviewer #2, comment 3: “Myalgic encephalomyelitis” should be listed among the search terms, as should MECFS, ME/CFS, CFSME, and other iterations. “Magnetic resonance spectroscopy” and related iterations should also be included.

Authors Response: Considered and modified in the manuscript. Fatigue Syndrome, Chronic is the mesh term and with it contains all associated search terms including Myalgic encephalomyelitis (S2 contains the full search term code). ME/CFS and other abbreviations have been omitted as myalgic encephalomyelitis/ Chronic Fatigue Syndrome are not accepted abbreviations and will always be introduced in its full form in the title and abstract. Magnetic resonance spectroscopy has been added as a search term. This has added seven studies that were previously not included: Chaudhuri, 2003; Mathew, 2009; Mueller, 2019; Murrough, 2010; Puri 2002; Shan, 2018A; Van der Schaaf, 2018. 

Reviewer #2, comment 4: The current draft reads like a review of neuroimaging studies - a very valuable resource - but not a review cognitive studies. Given that the focus is on neuroimaging per se and not cognitive dysfunction, “Cognitive” should probably not be in the title: for example, Gotts et al (2015) PLoS ONE is a study of sleep and cognition - two topics that are covered in this review. But it is not referenced or discussed in the current review because it was not a neuroimaging study. “A systematic review of neuroimaging….” may be an example of better phrasing for the title. If there is to be a focus on cognition per se, even outside of the scanner, then there should be an approach more like Cvejic et al. (2016), including search terms related to cognitive testing.

Authors Response: Considered and modified in the manuscript. The focus of this review is neurological changes as measured by neuroimaging techniques. Cognitive changes has been removed from the title and it now reads as: A Systematic Review of Neurological impairments in Myalgic Encephalomyelitis/ Chronic Fatigue Syndrome using neuroimaging techniques.

Reviewer #2, comment 5: The paper 'Costa et al (1995) Brainstem perfusion...' appears to have been missed; it used “SPET” single photon emission tomography, this may have been missed by using the more specific and constrained search of “tomography, emission-computed, single-photon” - perhaps simply using “tomography” would be more inclusive and it would also catch PET studies.

Authors Response: Considered and unmodified in the manuscript. 

Costa et al (1995) was identified in the search, however, was later excluded since using an alternative criterion than those mentioned in our inclusion/exclusion criteria (Oxford). The paper also included participants under the age of 18. 

Reviewer #2, comment 6: We suggest that the authors include “magnetic resonance spectroscopy” in the search terms to make the study list more inclusive. For example, Murrough et al. (2010), NMR in Biomedicine, Mathew et al. (2008), NMR in Biomedicine. From the 10 spectroscopy papers listed in VanElzakker, Brumfield, & Mejia 2019, only 2 were included in the current review. There have been a handful of neuroimaging reviews that could also be a source for appropriate papers, by looking through references:

Morris, Berk & Puri (2018); VanElzakker, Brumfield, & Mejia 2019;

Authors Response: Considered and modified in the manuscript. As mentioned previously, magnetic resonance spectrometry has been added as a search term: this has allowed the inclusion of additional papers including Murrough et al. (2010). The reference was checked and all mentioned references were identified in the search but were later excluded on the basis of inclusion/ exclusion criteria. Yoshiuchi (2006) and Lange (2001): separated the ME/CFS patients based on having a diagnosis of depression when that is an exclusion criterion for diagnosis. Costa (1995) was excluded due to the inclusion of the paediatric population. Peterson (1994) and Ichise (1992) were published prior to the establishment of the Fukuda criteria (1994). Brooks (2000) and Tamada (2000) used an alternative Holmes (1988) criteria. 

Reviewer #2, comment 7: We were surprised that a vast majority of studies were based on Fukuda criteria. One key problem with the Fukuda definition is that it likely includes individuals who deal with fatigue but do not share the same pathophysiology as ‘true’ ME/CFS. Inclusion of “Infectious Onset” as a ME/CFS cohort descriptor would be particularly helpful, if this is not possible it should be described as a limitation.

Authors Response: Considered and modified in the manuscript. ‘infectious onset’ information was not available in the studies. A statement in the discussion section has been added to address this: due to this potential overlap, the inclusion of additional patient characteristic information such as infection onset is also an important consideration for future studies (line 417).

Reviewer #2, comment 8: Line 68: Please correct this numbering for the supplementary material (listed as S1 but is actually S2). This problem occurs throughout the paper. Please address all mentions of supplementary materials.

Authors Response: Considered and modified in the manuscript. All supplementary material has been correctly numbered in the revised manuscript. 

Reviewer #2, comment 9: Line 85: IOM criteria not listed here; given its prominence, if this was deliberate it should be stated explicitly. This is especially salient given the preponderance of Fukuda papers referenced.

Authors Response: Considered and partially modified in the manuscript. No articles identified in this search used the IOM criteria. The IOM criteria in research hasn’t been as well established as the other criterions in research. The main text has been edited to include a statement that IOM is part of the inclusion criteria “diagnosis of ME/CFS used the following case criteria: Fukuda (1994), CCC (2003), or ICC (2011) or the Institute of Medicine (IOM) diagnostic criteria (2015)” but no studies used this criteria, this was also explicitly stated in the manuscript “No papers utilised the IOM diagnostic criteria (line 135).” 

Reviewer #2, comment 10: The authors describe that this manuscript intended to review and survey studies comparing ME/CFS to healthy controls, but it appears that some papers which make such a comparison were not included simply because they also made other comparisons. There are some examples of papers that have more than two groups - for example, Natelson et al (2017), Fatigue Biomedicine Health & Behavior 5(1). Given that there was a comparison between ME/CFS and healthy controls in that paper, it should probably be included (and the comparison to fibromyalgia can be ignored, in accordance with the self-imposed criteria for the current review). Other examples of this should be looked for.

Authors Response: Considered and unmodified in the manuscript. Selected papers that were not included such as Natelson et al (2017) were excluded for other reasons such as not including an age descriptor. Murrough et al (2010) was included as it complied with all inclusion criteria. 

Reviewer #2, comment 11: Please explain the reasoning for excluding sleep intervention studies. Was it all intervention studies that appeared in your search that you excluded or solely the sleep intervention studies? Please explain if this distinction was made and why. Was the exclusion only for studies that attempted to alter or influence sleep?

Authors Response: Considered and partially modified in the manuscript. All intervention studies were excluded from the review. There are several articles that describe the influence intervention regimes have on neuroimaging findings in ME/CFS patients compared to HCs that justifies its own investigation and was beyond the intended focus of this review. Initially, Perrin et al. 2010 that involved a selected group receiving osteopathic treatment and another group receiving any treatment of their choice was initially included in the study. This study has now been removed from the analysis.

Reviewer #2, comment 12: Line 113: the PubMed search term list was unusually short compared to the others, please explain.

Authors Response: Considered and unmodified in the manuscript. PubMed has a mesh database that allows expansion of search terms to include all related variants of the term. The other databases did not have this option, so the code was adjusted to include all the additional search terms that PubMed’s mesh database captures. 

Reviewer #2, comment 13: Line 128, did any studies report race or ethnicity?

Authors Response: Considered and modified in the manuscript. A description on studies which reported race or ethnicity has now been provided. This statement reads as: There were only 10 studies which reported race, where majority of the participants were Caucasian (line 128).

Reviewer #2, comment 14: Line 136, in general, sentences should not begin with numerals (see elsewhere as well, e.g., Line 121) - a minor formatting issue throughout that should be edited. Line 328 vs line 329, “Table” and “Figure” should have consistent capitalization throughout.

Authors Response: Considered and modified in the manuscript. The whole manuscript has been checked for sentences that begin with numbers and has been adjusted. The figures also now have consistent capitalisation throughout the text. To comply with PLOS ONE formatting, tables have changed to supporting information and have been relabelled accordingly. 

Reviewer #2, comment 15: We believe the authors meant “spectroscopy” - this is technically an fMRI technique (functional). In general, neuroimaging studies are usually described according to a distinction between functional and structural; this distinction could be made clear throughout the paper.

Authors Response: Considered and modified in the manuscript. The line has been adjusted to say spectroscopy (line 152). 

Reviewer #2, comment 16: Line 147 describes MRS and finding a difference in rCBF, but this is not something that MRS can do. We believe the authors intended to describe the ASL technique’s finding. In general, there were several instances in which more than one technique was used (MRI machines can of course do multiple methods of structure and function) and a result was described according to a different method. ASL is usually considered a form of fMRI (it describes function, not structure).

Authors Response: Considered and modified in the manuscript. The line stating that MRS was used to measure rCBF has been adjusted to say ASL. The complete main text has been cross-checked to ensure that the correct technique is being described (line 161). The tables have also been adjusted to include all techniques.

Reviewer #2, comment 17: Line 163: Did the authors intend to write “intercranial compliance” ? In general, this is an example of a finding that would benefit from even a brief interpretation or explanation.

Authors Response: Considered and modified in the manuscript. “Intercranial compliance” has been adjusted to “intracranial compliance”. The equation of intracranial compliance has been provided to add context: “Severity of OI symptoms were associated with lower intracranial compliance (change in volume per unit change in pressure)” (line 164).

Reviewer #2, comment 18: Line 187 - (organisation) hippocampus function is listed in a section on brain structure, and EEG findings are not listed in the section on electrical activity.

Authors Response: Considered and modified in the manuscript. The brain structure and function section has now been merged (line 167). The EEG findings have been included in the section on electrical activity, it reads as: a twin study that used EEG Low-resolution electromagnetic tomography analysis (LORETA) found that there was significantly higher delta power in the left uncus and parahippocampal gyrus and higher theta power in the cingulate cortex and right superior frontal gyrus in the ME/CFS monozygotic twin compared with the healthy twin (line 262). 

Reviewer #2, comment 19: Line 189 - no comma after “Significantly”

Authors Response: Considered and modified in the manuscript. The comma has been removed. 

Reviewer #2, comment 20: Line 200 - It is true that myelin levels may be associated with T1 signal; however, studies examining myelin usually use a T1/T2 ratio and not simply T1 signal, which is much harder to interpret. It should be made clear that “myelin levels” is an interpretation of a finding as opposed to an empirical finding (elsewhere in the paper as well). Relatedly, Line 203 should explicitly state what variables were correlated, and take care to describe the difference between the reported statistical finding and its proposed interpretation.

Authors Response: Considered and modified in the manuscript. Emphasis on the difference between interpretations and empirical findings were made throughout the text and all interpretations have been removed from the results section. 

Reviewer #2, comment 21: Line 207 - the word “study” should follow “DTI”, line 208 the term FA appears unnecessarily; the word “arcuate” is misspelled

Authors Response: Considered and modified in the manuscript. This paragraph was removed from the paper. 

Reviewer #2, comment 22: Line 213-220 - terms like “decrease” used as a verb sounds like longitudinal study findings; this wording should be clarified in case-control comparisons that cannot delineate a pre-existing risk factor from a disease process (i.e., smaller volumes may be the cause or may be the effect of a disease)

Authors Response: Considered and modified in the manuscript. The use of the term “decrease” has been switched in case-control comparisons to words such as “less” to differentiate between longitudinal studies throughout the whole manuscript. 

Reviewer #2, comment 23: Line 257 - Another example of a reported finding that would benefit from even a small amount of explanation, example wording could be something like “small worldness, or propensity for local functional connectivity over long-range functional connectivity...” Line 264, Citation 46 - this is another example of why contextualization is important: the “inhibitory state” was an interpretation of results in a study comparing 24 patients to 23 controls, and was not an empirical finding. Again, the sleep & electrical sections could benefit from a small overall summary so that it is not simply a list of papers.

Authors Response: Considered and partially modified in the manuscript. An explanation of small worldness has been provided: In another investigation small worldness, defined by the level of integration and clustering of networks, in the delta band was found to be significantly lower in ME/CFS patients compared with HCs (line 268). The sentence regarding inhibitory state has been removed. It is difficult to compose an overall comparison because there is minimal overlap between the studies conducted and the methodologies used.

Reviewer #2, comment 24: Line 291 - patients had or patients exhibited; not patients reported

Authors Response: Considered and modified in the manuscript. The text has been adjusted to “patients exhibited” instead of patients reported. 

Reviewer #2, comment 25: Line 302 unclear wording, line 309 unclear wording; but more importantly: line 302 vs line 308 different auditory tasks appear to have a positive vs a null finding. Are the two studies comparable? Are they inconsistent? Is there a possible conceptual or mechanistic reason for why these two studies seemed to find different things? An example where contextualization would be helpful.

Authors Response: Considered and modified in the manuscript. 

Reviewer #2, comment 26: Line 312 - dACC is usually not referred to as a nucleus

Authors Response: Considered and modified in the manuscript. “Dorsal anterior cingulate nucleus” was adjusted to say “dorsal anterior cingulate cortex” instead (line 177). 

Reviewer #2, comment 27: Line 333 - briefly define ‘role function’ - also wording in 353 - 356, explain ‘exposure period was sufficient to’

Authors Response: Considered and modified in the manuscript. Literature was searched on role function; however, a definition of the term has not been provided. Due to this, this was removed from the manuscript. “Exposure period was sufficient to” was changed to “the exposure period was sufficient to show an effect” to provide more context (line 371). 

Reviewer #2, comment 28: Line 337 - Organisation: Quality assessment seems more like a “Methods” topic, used in the initial selection of papers. In general, one cannot read this section and understand what the authors did - if these methods are truly central to paper quality assessment, this table should either be moved from supplemental into the main paper, or the text should be much more descriptive and not rely upon flipping back and forth between the text and the table to understand what is being described. Line 523 begins a clearer description of what seems to be described in 337.

Authors Response: Considered and modified in the manuscript. Greater description has now been provided in the methods section: Each checklist item assesses the following: (1) group matching, (2) source population, (3) criteria, (4) method of exposure, (5) assessment of exposure, (6) identification of confounding variables, (7) management of confounding variables, (8) measurement of outcomes, (9) exposure period selection, (10) statistical analysis (line 111). Details of the methodology has also been removed from the discussion section. Upon consultation of guidelines and correspondence with a PLOS ONE editorial team member (31st March correspondence) the quality assessment table is required to be presented as supporting information and can’t be moved to the main text. 

Reviewer #2, comment 29: Line 431, references should follow each statement separately - in other words, studies that reported white matter hyperintensities should follow “WM hyperintensities” - white matter atrophy and white matter volume are synonymous, but references supporting this statement should follow that statement specifically.

Authors Response: Considered and modified in the manuscript. The references have been adjusted accordingly. 

Reviewer #2, comment 30: Line 432 - “not of pathological significance” is an interpretation. Line 438 “established” is too strong; the author is making an argument based upon interpretation of data (this also directly contradicts line 482); 444 - “upregulation of myelin” is an interpretation and not an empirical finding.

Authors Response: Considered and modified in the manuscript. The statement “not of pathological significance” has been removed. Established has been adjusted to proposed. The statement on the upregulation of myelin has been framed to show it is an interpretation and not an empirical finding: It was suggested that this may be due to an upregulation of myelin in the sensorimotor cortex which was a potential compensatory mechanism to maintain brainstem-sensorimotor connectivity (line 456).

Reviewer #2, comment 31: Line 486 - define ICN, and please make sure all other acronyms are defined

Authors Response: Considered and modified in the manuscript. ICN was only mentioned once and did not need to be abbreviated it is presented in full form (line 492). The manuscript has been thoroughly checked to ensure all acronyms have been defined. 

Reviewer #2, comment 32: Line 487 - This sentence fragment seems like evidence that the paper was not proofread by all authors. Please assure that all authors have read and edited the paper before resubmission.

Authors Response: Considered and modified in the manuscript. The sentence fragment has been corrected. The paper has been read and edited by all authors.

Reviewer #2, comment 33: Line 492 - hippocampus is itself a limbic structure. Was 'hippocampus' and 'hypothalamus' again switched? One does not normally think of hippocampus as central to arousal.

Authors Response: Considered and modified in the manuscript. The author was referring to hippocampus, however, this sentence has now been removed from the manuscript. 

Reviewer #2, comment 34: Line 518 - “compensation” is an example of an interpretation or theory, not an empirical finding, and should be framed as such.

Authors Response: Considered and modified in the manuscript. This statement has been rephrased to frame it as an interpretation or theory not an empirical finding: Increases in FC in patient groups in response to disrupted FC elsewhere is another possible example of compensation (line 516).

Reviewer #2, comment 35: Table: For the table, the term USA is more appropriate, because Brazil is in “America” but this does not seem like important information when judging the quality and findings of studies. Perhaps a “Method of analysis” column has room for more information, for example it could include the Tesla strength of MRI. The “sample type” column did not seem important given that all entries said “CFS/ME.” The table also lists all studies as “prospective” but we believe this may be a misapplication of the term, which would mean that subjects were scanned before they developed the disease (see also Line 122, 123, 124). http://sphweb.bumc.bu.edu/otlt/MPH-Modules/EP/EP713_AnalyticOverview/EP713_AnalyticOverview3.html

It is again important that senior authors carefully read the entire paper.

Authors Response: Considered and modified in the manuscript. The country of origin of the study section on the table has been removed as It is not important information to deduce quality and findings of the studies. The sample type column has also been removed. The tesla strength of MRI has been added to the method of analysis section. The word “prospective” was removed from the study design section. 

Reviewer #3: 

Reviewer #3, comment 1: Why was Fukuda the primary chosen criteria? As the authors have rightly pointed out, it is too broad according to the majority of researchers and clinicians.

Authors Response: Considered and partially modified in the manuscript. Fukuda was initially the internationally recognised case definition for ME/CFS therefore a large proportion of studies were based on this criterion. The CCC is now the internationally recognised case definition and there may be a shift in publications with more articles relying on this definition in the future.

Reviewer #3, comment 2: Are there no articles available using some of the more restrictive criteria (IOM for example?) that the authors could have used?

Authors Response: Considered and unmodified in the manuscript. Four different databases were searched and out of all the included papers none used the more restrictive criteria such as the ICC or IOM.

---

## [Decision Letter · Decision Letter 1]

8 Apr 2020

PONE-D-19-34471R1

A systematic review of neurological impairments in myalgic encephalomyelitis/ Chronic Fatigue Syndrome using neuroimaging techniques.

PLOS ONE

Dear Miss Maksoud,

Thank you for submitting your manuscript to PLOS ONE. After careful consideration, we feel that it has merit but does not fully meet PLOS ONE’s publication criteria as it currently stands. Therefore, we invite you to submit a revised version of the manuscript that addresses the points raised during the review process.

I would recommend authors carefully explain their inclusion criteria. In particular, authors should discuss how the application of different inclusion criteria could have affected their findings.

We would appreciate receiving your revised manuscript by May 23 2020 11:59PM. To enhance the reproducibility of your results, we recommend that if applicable you deposit your laboratory protocols in protocols.io, where a protocol can be assigned its own identifier (DOI) such that it can be cited independently in the future. For instructions see: http://journals.plos.org/plosone/s/submission-guidelines#loc-laboratory-protocols

We look forward to receiving your revised manuscript.

Kind regards,

Marcello Moccia

Academic Editor

PLOS ONE

Additional Editor Comments (if provided):

I would recommend authors carefully explain their inclusion criteria. In particular, authors should discuss how the application of different inclusion criteria could have affected their findings.

Reviewers' comments:

Reviewer's Responses to Questions

**Comments to the Author**

1. If the authors have adequately addressed your comments raised in a previous round of review and you feel that this manuscript is now acceptable for publication, you may indicate that here to bypass the “Comments to the Author” section, enter your conflict of interest statement in the “Confidential to Editor” section, and submit your "Accept" recommendation.

Reviewer #1: (No Response)

Reviewer #3: (No Response)

2. Is the manuscript technically sound, and do the data support the conclusions?

Reviewer #1: Yes

Reviewer #3: Yes

3. Has the statistical analysis been performed appropriately and rigorously? 

Reviewer #1: N/A

Reviewer #3: Yes

4. Have the authors made all data underlying the findings in their manuscript fully available?

Reviewer #1: Yes

Reviewer #3: Yes

5. Is the manuscript presented in an intelligible fashion and written in standard English?

Reviewer #1: Yes

Reviewer #3: Yes

6. Review Comments to the Author

Reviewer #1: This reviewer highly values the authors's efforts to modify the manuscript.

There is one point which are not completely satisfied.

The authors wrote to comment 1: "Kimura et al, was excluded during the

abstract/ title screening process as it did not contain any of the listed key words. "

This is not wrong, however, because this paper seems to be within the scope of this manuscript with regards to its content, the reviewer is still not completely convinced why this paper is excluded from this review. Please explain this point.

Reviewer #3: Fukuda criteria is way too broad, and such a review should have addressed some articles with more restrictive criteria. There is a strong lack of consensus among researchers and clinicians on it's reliability as far as specificity is concerned.

7. PLOS authors have the option to publish the peer review history of their article (what does this mean?). If published, this will include your full peer review and any attached files.

Reviewer #1: No

Reviewer #3: No

---

## [Author Response · Author response to Decision Letter 1]

15 Apr 2020

Reviewer #1, comment 1: The authors wrote to comment 1: "Kimura et al, was excluded during the abstract/ title screening process as it did not contain any of the listed key words. "

This is not wrong, however, because this paper seems to be within the scope of this manuscript with regards to its content, the reviewer is still not completely convinced why this paper is excluded from this review. Please explain this point.

Authors Response: Considered and modified in the manuscript. Our systematic review utilised standardised and internationally accepted PRISMA and Cochrane review guidelines (https://training.cochrane.org/handbook/current/chapter-04#section-4-4). This approach limits the introduction of bias from self-selection of papers. The inclusion of Kimura et al although within the scope of our review would be introducing a paper outside the set international guidelines and stringent inclusion criteria and may be a source of potential selection bias. The discussion has been modified to include descriptions of the paper’s findings (line 494). 

Reviewer #3: 

Reviewer #3, comment 1: Reviewer #3: Fukuda criteria is way too broad, and such a review should have addressed some articles with more restrictive criteria. There is a strong lack of consensus among researchers and clinicians on it's reliability as far as specificity is concerned.

Authors Response: Considered and modified in the manuscript. A statement in the discussion was provided regarding the limitation of including the Fukuda criteria in our search strategy: “Inclusion of the Fukuda criteria was a limitation of our search strategy, however, due to this criteria being still abundantly used in research and clinical practice, limiting our search to not include papers that relied on the Fukuda criteria would drastically reduce the availability of appropriate papers in this field.” (line 423)

---

## [Editor Report · Decision Letter 2]

16 Apr 2020

A systematic review of neurological impairments in myalgic encephalomyelitis/ Chronic Fatigue Syndrome using neuroimaging techniques.

PONE-D-19-34471R2

Dear Dr. Maksoud,

We are pleased to inform you that your manuscript has been judged scientifically suitable for publication and will be formally accepted for publication once it complies with all outstanding technical requirements.

With kind regards,

Marcello Moccia

Academic Editor

PLOS ONE

Additional Editor Comments (optional):

I wish to congratulate the authors for their manuscript.
---

## [Editor Report · Acceptance letter]

22 Apr 2020

PONE-D-19-34471R2 

A systematic review of neurological impairments in myalgic encephalomyelitis/ Chronic Fatigue Syndrome using neuroimaging techniques. 

Dear Dr. Maksoud:

I am pleased to inform you that your manuscript has been deemed suitable for publication in PLOS ONE. Congratulations! Your manuscript is now with our production department. 

With kind regards,

on behalf of

Dr. Marcello Moccia 

Academic Editor

PLOS ONE